# Sustainable Feedbacks of Colombian Paramos Involving Livestock, Agricultural Activities, and Sustainable Development Goals of the Agenda 2030

**Raúl Andrés Molina Benavides [1],* , Rómulo Campos Gaona [1], Hugo Sánchez Guerrero [1], Leonidas Giraldo Patiño [1] and Alberto Stanislao Atzori [2]**

[1] Grupo de investigación "Conservación, Mejoramiento y Utilización del Ganado Criollo Hartón del Valle y otros Recursos Genéticos Animales en el Sur-Occidente colombiano", Universidad Nacional de Colombia, Palmira 763531, Colombia; rcamposg@unal.edu.co (R.C.G.); hsanchezgu@unal.edu.co (H.S.G.); lgiraldopa@unal.edu.co (L.G.)

[2] Section of Animal Science, Department of Agriculture, Università degli Studi di Sassari, 07100 Sassari, Italy; asatzori@uniss.it

* Correspondence: ramolinab@unal.edu.co

**Abstract:** Colombian mountain Páramos are considered natural areas with a very important role for human life. Páramos provide, both in mountain and lowland areas, a multitude of ecosystem services which start from vegetation to soil sustainability. The sustainability of Páramos is however impaired by several anthropogenic activities, including agricultural and livestock practices. A system thinking approach was applied in this work to improve the systemic understanding of factors affecting sustainability and resilience of Páramos agro-ecosystems. Interdisciplinary literature evidences were summarized and conceptually analyzed in order to develop causal loop diagrams of Páramo system structures allowing describing the main feedback loops involving (involved in/connecting) the Páramo ecosystem and driving its sustainability. From the causal diagram analysis few insights to maintain the human presence in Páramos arose. The system analysis highlights that human presence in Páramos should be stimulated, avoiding agriculture and livestock activities as the main income source. Particularly, social interactions, education on the Páramos environmental and relevance of agricultural practices to foster ecosystem services and multiple rentable economic activities should be enhanced. The study also includes the role of the government in providing the Páramo inhabitants with payments for ecosystem services and environmental education aimed to boost sustainability. Sustainable Páramo management will apply specific leverages on the system to reach Sustainable Development Goals 6 (water), 8 (economic growth, employment and work), 13 (climate change), and 15 (sustainable use of terrestrial ecosystems) of the Agenda 2030.

**Keywords:** biodiversity; carbon; nutrient cycling; system dynamics; water

## 1. Introduction

In South America, the tropical Andes offer a series of ecosystems with different climates, diversity of soils, flora, fauna, and landscapes that allow their inhabitants to carry out heterogeneous productive activities [1–4]. The high mountain ecosystems, located between the upper limit of the Andean forest, and the lower limit of the glaciers or perpetual snow, are called Páramos [5]. These ecosystems are found in countries such as Colombia, Ecuador, Peru, Venezuela, and Costa Rica [6].

Colombia, with an area close to 3% of its continental area, represents 50% of the Páramos of the world classified, according to bio-geographical criteria, in 26 units ranked in sectors, districts, and complexes [5]. Many authors agreed with the difficult to uniquely define Páramos for its ecological

complexity and the heterogeneous factors that are embedded in this definition [5,7]. These ecosystems are generally located above 3000 m of altitude [2], however it is not easy to define the Páramo limits with a precise indication of altitude, geographical lines, or vegetation because the passage from forest to Páramos in high lands is identifiable with a gradual change in vegetation and soils from sub-Páramos to super Páramos (Figure 1; [8]).

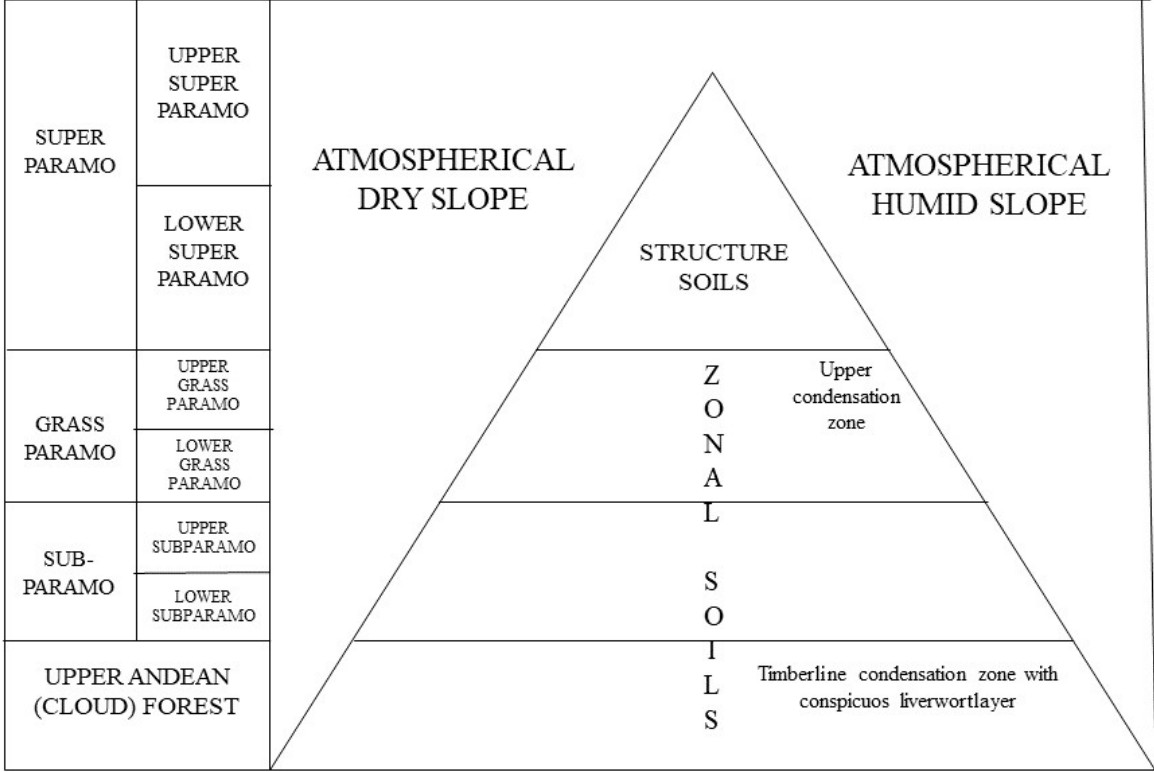

**Figure 1.** Ecological description of Páramo zonations based on altitude, soils, and vegetation (adapted from [8]).

Páramos are important mainly for their biological uniqueness and the ecosystem services (ES) they provide to human populations [9]. Among the ES offered by the Páramos we can find biological regulation (water regulation, microclimatic regulation, water purification, air purification, among others), provision (food for livestock, food for agriculture, fishing, water, among others), and cultural services (ecotourism, environmental education, spiritual, among others) [10]. It is intuitive to understand the ES role and the direct and indirect contributions that ecosystems offer to human beings for their well-being [11]. In this sense Páramos are considered as strategic ecosystems both for rural population living in the Páramo and for urban and industrial populations living far away. Most of the studies on Páramos described and partially quantified specific characteristics of Páramos area including ecological and/or socioeconomic aspects. It seems quite clear that is not possible to describe Páramos using only vegetation and altitude characteristics, but they also have to be considered a socioecosystem with biogeophysical, ecological, socioeconomic and cultural factors [5]. Despite the variety of ES offered by Páramos, all the studies highlights two main features representing the central axis of their environmental role: (i) Water: management of water resources through collection, regulation, and supply [12–14], (ii) Carbon: storage and uptake of atmospheric carbon through the retention of organic matter in the soil [1,15,16]. Looking at the perception of the local populations the role of water provision and food production are the most relevant ones [17].

The Páramos evolution took place in constant interaction with human populations throughout history, and their current state is the result of this process [18]. Human activity in the Páramos is

supposed to have existed for ten thousand years, mainly with religious and ceremonial purposes until the XIII century. Then, after the XVI century, an increase in livestock grazing and agricultural activities such as cereal and vegetable cultivation was observed [19]. Under anthropic pressures on land use, these ecosystems have been transformed into agroecosystems where activities such as agriculture, livestock, mining, extraction of natural resources, and tourism are carried out [3,15,20]. The anthropic activities on Páramos are nowadays mainly based on the economic interest of exploiting resources to increase incomes of families and neighborhoods. It has been clearly demonstrated that disturbing actions such as mining practices, livestock, and agriculture, especially if associated to fire practices, impair the hydrological and ecological sustainability of Páramo ecosystems [21].

The current management of the Páramos has to be subordinated to environmental sustainability issues towards two main approaches: (i) the importance of the management of the Páramo for its own equilibrium and (ii) the importance of the Páramo in the productive ecosystems of the valleys located within the watershed basin of the same Páramo. Nowadays, it appears quite clear that ecological and social functions of the Páramos are interdependent, that the boundaries of the Páramo ecosystems are larger than the limits of the Páramo areas, that targets of pure conservation are not among the winning strategy to face sustainability and also that climate change will affect the characteristics and the socioecological rules of the Páramo environment [22].

In this sense, a systemic view appears necessary to improve the understanding of the complexity and the formulation of management policies. In fact, a mere description of elements characterizing the Páramo agro-ecosystems is not sufficient to understand and identify the driving forces of future sustainability. It is necessary to define and highlight the interactions between elements and the emerging processes that drive the system behavior and the expected future changes. Few studies often indicate and list the need of future policies and recommend possible sustainable practices that can help to maintain the ecological equilibrium of Páramos [2,4,5]. Nevertheless, when problems are narrowly focused, there is an increased risk to generate delayed side effects that might impair the equilibrium of the whole system [17]. A systemic view and approach allows integrating different elements and feedbacks and broadens the discussion in terms of future effects and expected behaviors. Systemic thinking recognizes that systems are complex, integrated, and self-sustained. Small and large shocks will propagate through it, and impact other variables instantaneously or over time, since they are connected in some way [23]. The systemic questions related to Páramos could be summarized as follows: which effects do the anthropogenic activities in the Páramos have on the sustainability of nearby agroecosystems? Which feedbacks that involve the Páramos and the nearby agroecosystems are fundamental to achieve sustainability in mountains and plains from an ecological and socio-economic point of view? It will require a change in perspective while looking at the current biophysical and economic dynamics and a substantial increase of organizational strategies to enhance sustainable territorial management.

This work aimed at summarizing the complexity of the Páramo from a qualitative point of view with a preliminary critical analysis of the available literature by using a System Dynamics approach. A specific objective was to describe the relationship between fundamental biotic and anthropic elements that characterize the Páramo ecosystem in order to: (i) improve the understanding of the system structure and element functions; (ii) contribute to the understanding of the effects of anthropogenic activities on Páramo and valley agro-ecosystems; (iii) to contribute to the formulation of policies on the management, use and conservation of Páramos. The operative objective of the work was to review info from literature on specific aspects of Páramos characteristics and roles, connecting them in a systemic view and to develop a qualitative model using causal diagrams and systemic thinking techniques.

Finally, the elements form the systemic structure of Páramos will be associated with Sustainable Development Goals of Agenda 2030 (SDG) in order to assess a picture of the global role of Páramos in terms of sustainable development goals indicators.

## 2. Methods

### 2.1. Area of Study

The work was executed taking as main reference the Páramo ecosystems of Colombia, almost all located on the Andes mountain range (Figure 2). It is estimated that about 300,000 to 500,000 people are currently populating the Colombian Páramos [2]. Unfortunately, there is no specific census for the Páramo population, livestock consistency, and agricultural pressure. Available information usually refers to specific areas and locations and statistics are made at a local level to estimate population, cultivated land and crops, livestock species, and production systems [2]. From an environmental and ecological point of view the Páramo landscapes of the study area provide ecosystem services mainly related to biodiversity [15,24], collection and water supply [9,12] and among the anthropic activities typical of these Colombian ecosystems, the dual purpose cattle ranching [6,20], while ecotourism is also affirmed [14,25] (Figure 3). Economic activities of local products are mainly oriented towards local markets, to emergent markets in periods of increasing population and to city markets. Small producers bring their products directly to emergent and established markets of the cities or, alternatively they sell the products to buyers that collect the products of different producers to provide larger markets in the cities [2].

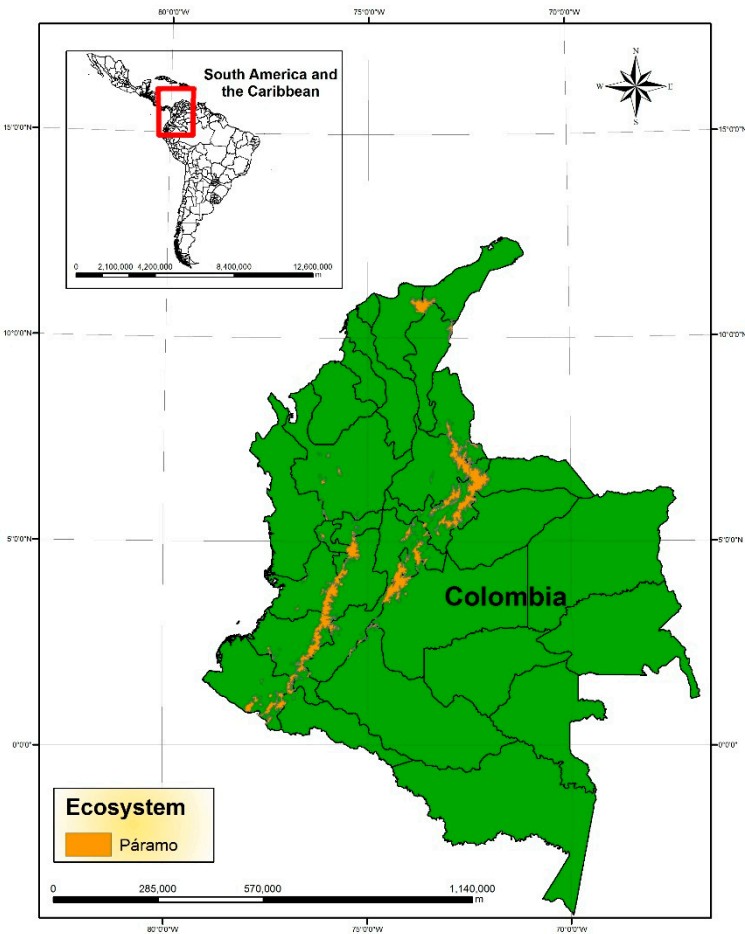

**Figure 2.** Geographic location of the different Páramo ecosystems of Colombia. Data source: elaboration of original figures from [5].

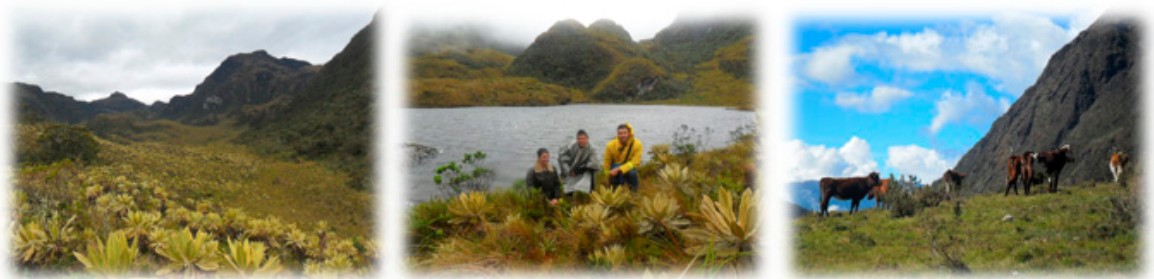

**Figure 3.** Examples of landscape, ecosystem services (collection, regulation and water supply, and ecotourism), and anthropic activity (livestock) representative of the Páramo ecosystems of Colombia. Páramo de las Hermosas, Palmira-Valle del Cauca.

### 2.2. Literature Review and Causal Loop Diagrams Development

The methodological approach used in this paper was driven by the idea to transfer the core of literature knowledge on Páramo in a causal diagram representing the systemic perspective of the Páramo ecosystem. In other terms to perform a preliminary review of Páramo studies using causal diagrams. The causal loop diagrams (CLD) are part of the mapping language proposed by the Systems Dynamics methodology, whose main purpose is to propose an approach to target the hidden core structure driving the behavior of the system [23,26–28]. Drawing causal maps or CLD using ST and SD approaches is a technique for mapping the existence of the feedback within and across interacting subsystems. The modeling process demands the identification and definition of problems, the overall system conceptualization, and the qualitative modeling, which often impact our initial system understanding. ST and SD can support designing better policies working on the systems structure both enhancing insights elicitation and increasing the stimulus to performing accurate policy discussion and evaluation [29]. Through the CLD, an explicit understanding of the problem can be generated, identifying the relationships between the components of the structure, especially the biotic and socioeconomic aspects. The agro-ecosystems studied in this paper can be considered as a set of human, physical, technological and animal resources, integrated from relationships and functions [28–30]. Several studies were carried out to formulate polices based on CLDs in agriculture and livestock systems [28–31] whereas any study was performed using a system thinking approach on Páramo equilibrium. CLD are composed of a set of symbols that represent the causal structure of a dynamic system. This set includes variables, arrows indicating causal relationships between them (with polarity "+" or "−", meaning mathematical correlation) and symbols that identify the feedback loops and their polarities (Reinforcement (R/+) or Balance (B/−)). The balance loops seek to regulate the system, the reinforcing loops favor the growth or decrease thereof [23,32,33].

A broad literature review was firstly performed to gather information on Páramos. The first search was performed on scientific journals databases using "Paramo" and "Colombia" keywords. A further deeper research was carried out adding livestock, agriculture, and water keywords in order to better focus on husbandry-related activities. A large literature review was also performed on literature in Spanish based on PhD theses, governmental reports, laws, informative articles, and other unofficial literature on Páramo characteristics, statistics, socioeconomic role and equilibrium management. The collected literature was considered exhaustive for the purposes of this paper which was not aimed at exploring all the details of the Páramo ecosystem, but to capture the most relevant elements that can characterize it in a systemic view. Materials obtained from various literature resources (articles, theses, books, informal literature) were then selected on the basis of their content in order to avoid redundancies and focus on the key messages of each work. Selected information sources where then classified on the basis of their domain in order to start the system characterization as reported for example in Table 1. Key messages from these selected sources were used to map the most important

biological and social interactions described in the literature and to further reduce the redundancy of information provided by the papers.

System boundaries included the concept of Páramo biological limits in highlands, the Páramo watersheds, and the lowland areas affected by the ecosystem services provided by Páramos and their inhabitants and activities.

A series of preliminary conceptual maps were then developed to describe features of Páramo ecosystems as reported in the literature for narrowed subsystems or domains. Conceptual maps are essentially causal loops diagrams [34]. The purpose of this conceptual modeling was to identify the variables and key interactions to improve the understanding of the studied system in terms of feedback cycles, delays, and leverage points that remarks the endogenous origin of system elements. In this work, the CLD were built finding connections among recognized key variables of the system and trying to "close the loop", meaning to track the nonlinearity and counterintuitive nature of the Páramo system in order to capture its latent structure [35]. The connections, within each loop, were drawn checking the characteristics of causality and polarity among them, double checking their roles in the system reference, and following evidences, concepts, and data published in the literature to warrant the correspondence between the CLD and the real reference system [23]. In this phase the authors worked in a group of two diagram developers and three diagram reviewers in order to have a sort of internal validation of fixed concepts, both in terms of SD methodological steps and Páramo knowledge and expertise. All the links, causal relationships, and polarities proposed in the developed CLD were double checked in respect to the description found in the literature [36]. Feedback loops of the CLD were identified and labeled targeting their influence on Páramo ecosystems, ecosystem services, and anthropic activities.

The preliminary diagrams did not explore the ecosystem domains in detail. The diagrams described reliable basic structures and feedbacks that were figured out from key messages of the considered studies. In a second step, the preliminary diagrams were merged and simplified in a single CLD in order to get a big picture of the Páramo system and to complete the modeling cycle with a much simpler and fundamental version of the working model [37]. A further validation step was conducted discussing the CLD with Páramo experts, especially researchers on the areas of climatology, geology, and socioeconomics from the National University of Colombia in Palmira.

**Table 1.** Examples of literature classification to the development of the casual loop diagram (CLD).

| Area | Journal | Focus | Author and Year |
|---|---|---|---|
| Environmental | Physical Geography | Soil | McKnight et al., 2017 [1] |
| Environmental | Ecological Engineering | Soil, agriculture, livestock | Otero et al., 2011 [12] |
| Environmental | Water Resources Research | Water | Poveda, Jaramillo, & Vallejo, 2014 [38]; Builes-Jaramillo & Poveda, 2018 [39] |
| Environmental | Applied Vegetation Science | Vegetation | Cabrera et al., 2018 [15] |
| Environmental | CEDAMAZ | Carbon, soil | Ayala, Villa, Aguirre, & Aguirre, 2014 [24] |
| Environmental, technical | Plant Ecology | Livestock, vegetation | Cierjacks, Ruhr, Wesche, & Hensen, 2008 [40] |
| Environmental, technical | Landscape Ecology | Agriculture, vegetation | Matson & Bart, 2013 [41] |
| Economic, environmental, social | Revista de investigación agraria y ambiental | Husbandry | Molina & Sánchez, 2017 [42] |
| Social | Food Policy | Food security | Burchi & De Muro, 2016 [43] |

**Table 1.** *Cont.*

| Area | Journal | Focus | Author and Year |
|---|---|---|---|
| Social | Agriculture | Food security in transition countries | Armendáriz, Armenia, & Atzori, 2016 [44] |
| Social, environmental | Annals of the American Association of Geographers | Water, payment for ecosystem services | Farley & Bremer, 2017 [9] |
| Social, environmental | World Development | Payment for ecosystem services | Hayes, Murtinho, & Wolff, 2017 [45] |
| Social, economic, environmental | Marine and Freshwater Research | Conservation, governance | Lynch, Kalumanga, & Ospina, 2016 [14] |
| Thecnical | Biotecnología en el Sector Agropecuario y Agroindustrial | agriculture | Ruiz et al., 2015 [20] |
| Thecnical | Revista U.D.C.A Actualidad & Divulgación Científica | Livestock | Bermudez, Arenas, & Moreno-Melo, 2017 [46] |
| Thecnical, environmental | Science of the Total Environment | Agriculture, livestock, soils | Avellaneda-Torres et al., 2018 [6] |

The CLD would reflect part of the complexity of the real system as representable in a mental model of the Colombian Páramos. Once the CLD was developed and consolidated, it was used to provide insights on possible sustainable policies as detailed below and to discuss the most recent governmental regulations and managerial policies of Colombian Páramos.

Finally, feedback loops generated in this work through the causal diagrams were associated to sustainable development goals (SDGs) proposed by the Agenda 2030 of United Nations with the aim to evidence which sustainable goals could be target by polices aimed to improve Páramo management and conservation.

## 3. Results

The causal diagrams present the relationship between Páramo ecosystems, their ecosystem services, their inhabitants, and the anthropic activities carried out within these zones. The effects on nearby areas affected by Páramo ecosystem services and products were also considered. A first diagram shows the variables associated with the high mountain population and the agricultural activities that guarantee their support (Figure 4).

The loops R1, B1 and B2 refer to the Páramos' flows of people, immigration, and emigration. This structure is typical of human population dynamics; if births (R1) are greater than deaths and emigration of the inhabitants (B1 and B2, respectively), then the population will increase, otherwise, the number of inhabitants may remain in dynamic equilibrium or decrease [32,33,47]. In these areas emigrations are frequent when resources for security and safety of the place decrease [31]. These incomers carry out agricultural activities (livestock and potato crops) [6,20,42] which allows them to generate food for their own consumption, contributing to their independence and food security [44,46,48], and to their nutrition and as a consequence, improving the reproduction rates of the population in a local scale (R2 and R3) [43,46]. On the other hand, the income generated by the sale of the surplus (milk, meat, potatoes) (R4 and R5), is highly important for the economy of the communities, and could contribute to the permanence of its inhabitants [46,47,49].

Following the system structure in the mental models, the variables related to the ecosystem services of the Páramos and their relationship with agricultural activities were also included in the causal diagram (Figure 5).

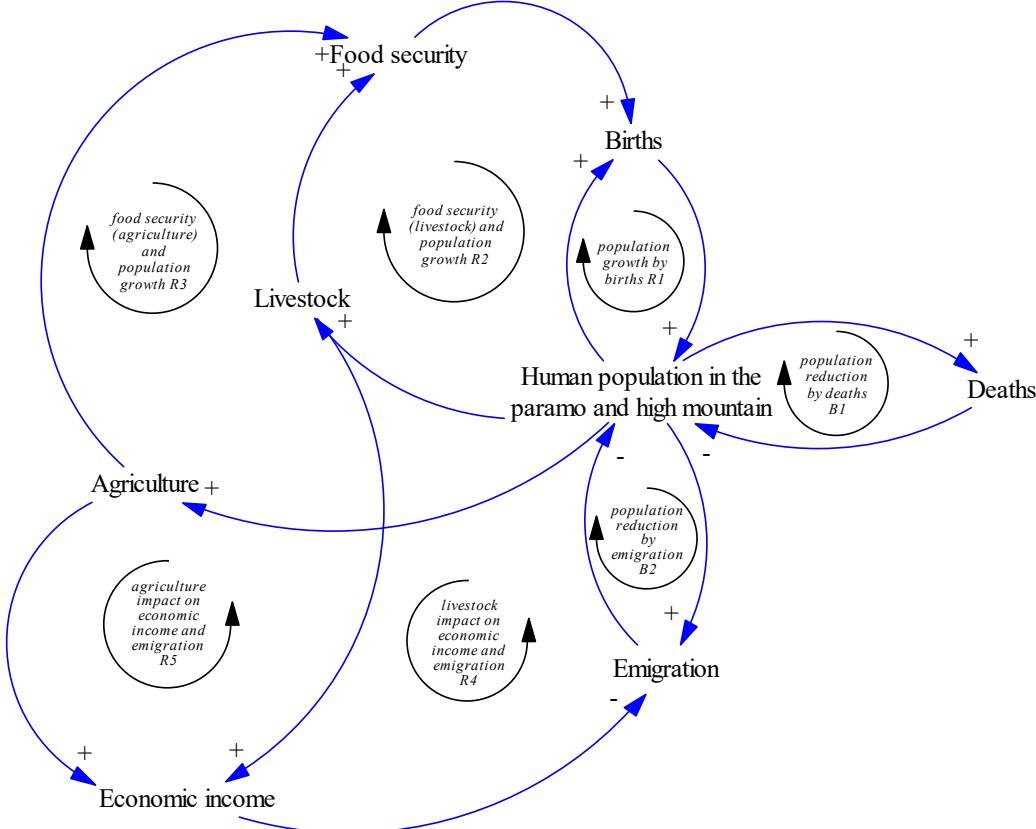

**Figure 4.** Causal loop diagram showing the relationship between the population dynamics of the Páramos and their agricultural activities.

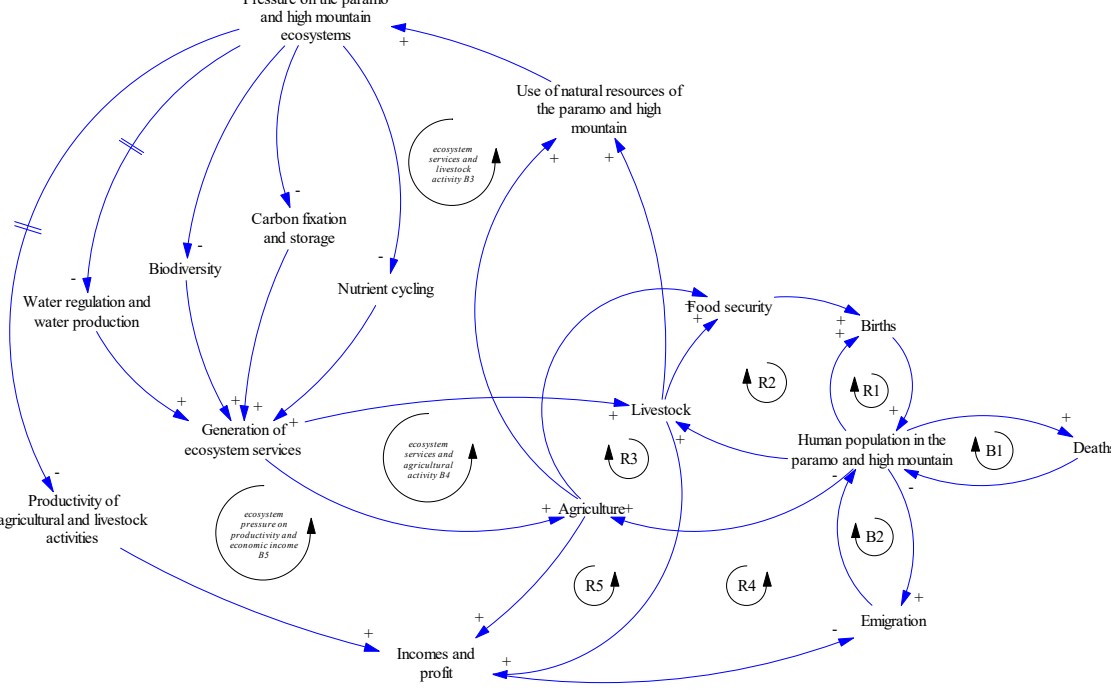

**Figure 5.** Causal loop diagram showing the relationship between the population dynamics of the Páramos and their agricultural activities.

Agricultural and livestock activities exert extractive pressure on the natural resources of Páramo ecosystems (soil, vegetation, water) [40,50–52]. For example, the trampling caused by cattle for long periods of time generates soil compaction [53–55] which in turn, results in less water infiltration, more runoff [56], reduction of plant cover [57], and lower absorption of nutrients by plants [58]. This negative effect on ES causes reductions in the productivity of agro-ecosystems [59,60], thus forming three negative feedback loops (B3, B4 and B5, Figure 5). The reduction of productivity is expected in the long term, due to a deterioration of the biochemical and hydrological equilibrium of the Páramo. The effects will not necessarily appear in the short term. The reduction in the income of the inhabitants lead people to migrate towards the cities in search of better labor opportunities [47,61]. Several authors also pointed out the called effects of "paramization", defined as the change in vegetal natural covers from forest to Páramo due to anthropic interventions of deforestation which determines land use change [21,62,63].

The existing connection between the migration of the Páramo inhabitants and their immigration to the cities has been demonstrated as a solution that ends generating social problems and food security (Figure 6). This migration of the rural population to the urban areas of the country does not generate the revenues that the peasants expect, on the contrary, many of the villagers who leave the countryside contribute to increasing misery and violence in the cities [64]. In addition to the low profitability, as a cause of migration of the inhabitants, the absence of the state (seen in the lack of basic services and infrastructure) [14,42,65] and the armed conflict [61], have promoted this phenomenon, bringing with it the displacement of people with a historical legacy in these territories. The absence of these people in the Páramos implies a series of negative consequences, for example, the entry of new people into these areas, without any traditional practical knowledge about the territory and its millenary management. Therefore, without the accompaniment of the natives, they can make incorrect uses of vegetation (cutting down trees, cutting leaves and typical vegetation like "frailejones"), increasing the amount of waste, causing fires, among others [66] and enhancing the already cited effect of paramization [21,63].

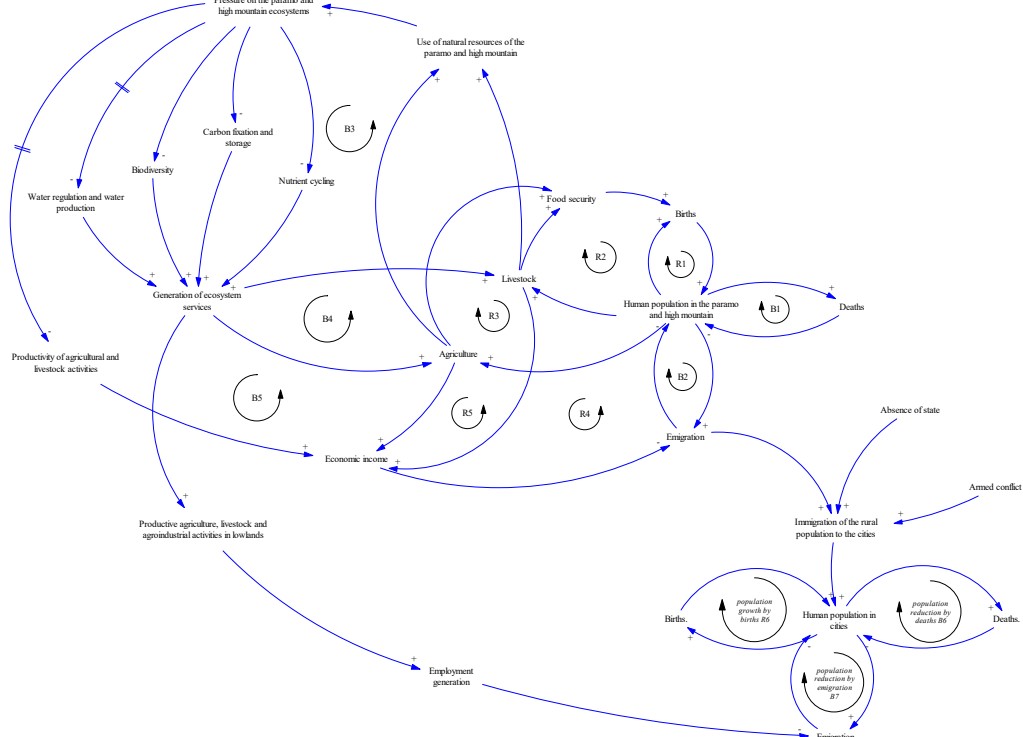

**Figure 6.** Causal loop diagram showing the benefit of ecosystem services to low-lying areas and the connection between the populations of the Páramos and the cities.

The ecosystem services generated by the Páramos also benefit the productive agricultural, livestock, and agro-industrial activities of the low areas (cities, municipalities, townships), generating employment, stability, and welfare for these populations [20] (Figure 6). However, the arrival of peasant settlers in urban areas can counteract this positive effect, contributing to the emigration of urban dwellers to other countries.

In addition, the ecosystem services generated by the Páramos, mainly through its water contribution, are indispensable for the economy of the country and the agricultural and industrial production of the lowlands, which allows the generation of employment, stability and welfare of these populations. [16,54]. However, the displacement of peasants to urban areas for different reasons (violence, lack of employment, basic services, schools, health services, among others) [28], trigger social dynamics marked by distrust and prevention, triggering conflicts of coexistence that in many cases generate the emigration of urban dwellers to other countries [55]. The population of Páramos and their neighbors could also be affected by climatic change and global warming caused by the increase of greenhouse gases in the atmosphere ($CO_2$, $CH_4$ and $N_2O$) [20,42,67]. It would mean that with global warming we could expect the displacement of life zones to their higher parts. On the other hand, if this vertical displacement is not possible at all, it could bring negative consequences on these ecosystems. This scenario of climate change, in many cases, would mean the reduction of the area of the Páramos, even to the point of disappearance [1,13,16]. For example, in the twentieth century small glaciers disappeared, and to continue the increase in environmental temperature, the snow peaks of Ruiz and Tolima would be the next to disappear [66]. Climate change is expected to create many changes in the Páramo ecological features and human activities [21].

*Intervention Policies*

The system thinking approach developed with causal diagrams allows both the evaluation of potential policies on the polarity of feedback loops and to qualitatively challenge the response of the system in the short and long term [34]. The policies proposed in this work included the role of the national government and institutional stakeholders in these territories which also are driven by the goals of Agenda 2030 and targeting sustainable development of the planet. The policies tested with the model presented are the following:

(1)   Environmental education
(2)   Payment for ecosystem services
(3)    Reforestation

The three polices were based on the governmental law N° 1930 (of 27 July 2018) which clearly indicated the guidelines and actions for the general management in Colombia. In particular, policy 1 was based on the current environmental education policy promulgated by the Ministry of National Education and the Ministry of Environment, Housing and Territorial Development, which establishes the systemic vision of the environment and the integral formation of the human being. Environmental education should be considered a crosswise axis in all contexts of the country. Policy 2 is supported by Decree 1007 of 14 June 2018, Ministry of Environment and Sustainable Development of Colombia, which aims to regulate the incentive for environmental payments. Environmental payments, (whether economic or in kind, recognized by the stakeholders of environmental services to owners, holders or occupants in good faith without fault for the preservation and restoration actions in strategic areas and ecosystems) are executed through the conclusion of voluntary agreements between those interested in environmental services and beneficiaries of the incentive. Policy 3 is given by the institutional framework for clean development mechanisms in Colombia, a commitment acquired in the United Nations framework convention on climate change. Figure 7 shows the effects of the proposed policies on the system equilibrium and on the affected variables and possible changes of polarity on the loops. The implementation of policies suggests that a strong presence of the government is fundamental in these territories. The CLD also showed the fundamental role of the government in the management of

Páramos (Figure 7). A leverage point could be represented by taxes paid by beneficiaries of ecosystem services in low-lying areas to promote, under governmental actions, programs for the sustainability of the Páramo ecosystems and the role of their inhabitants. Through these new variables included in the model, different positive feedback loops emerged and will stimulate the growth of the system [14,68].

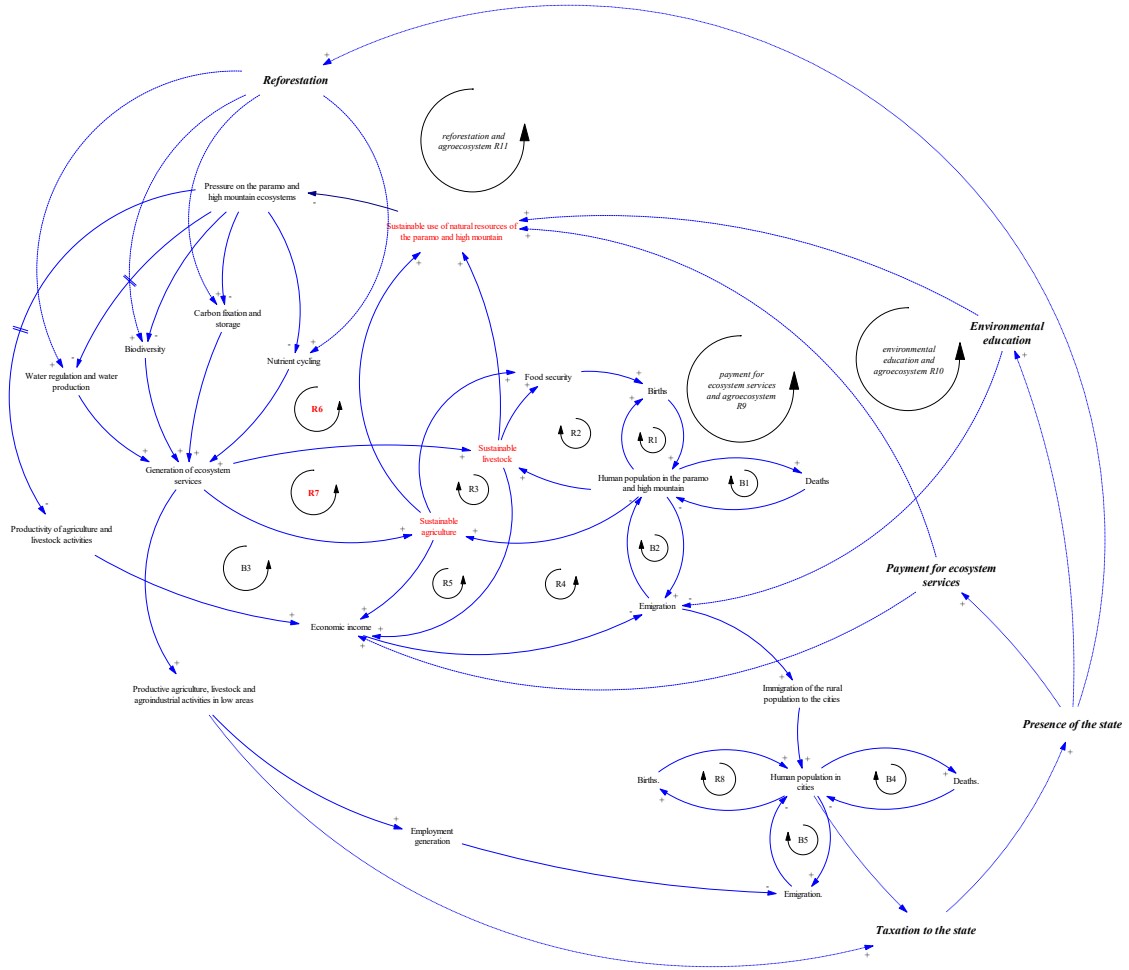

**Figure 7.** Causal loop diagram showing the effect of the proposed policies on Páramo populations, their agricultural activities and ecosystem services.

- Environmental education: Environmental education tends to increase the inhabitants' awareness on the relevance of environmental equilibrium on human life and also on the role of ecosystem services on sustainability. It is fundamental to have human beings with beneficial action on nature, making a sustainable use of the natural resources to which it has access [69,70]; In addition to education and education on the conservation of the Páramos, this policy should promote the continuity of Páramo inhabitants in their habitats, showing them the fundamental role they play in the process, fostering a sense of belonging for their territories. This policy will promote the sustainable use of Páramo ecosystems, including the anthropogenic activities practiced (agriculture, livestock, artisanal productions, and tourism), and will reduce the displacement of Páramo inhabitants to other zones (Figure 7).
- Payments for environmental services (PES): PES emerge as a conservation strategy that encourages owners, through economic and tax benefits, to take care and strengthen natural resources and ecosystem services related to water, carbon, and biodiversity [9,45]. This policy will promote

the conservation and sustainable use of the natural resources of the Páramo ecosystems and will become the main income of the inhabitants of these zones.

Both policies, environmental education and PES, seek to promote actions for the reconversion of livestock and agriculture to less impactful activities with natural resources, but that offer people livelihoods that ensure food security, health, education, and decent housing. In particular, to be sustainable and viable, the minimum values of PES should at least equal the benefits provided in ecological terms and in opportunity costs that should be sustained to provide the same services with specific actions and less expensive than the avoided damages of the entire ecosystem. The payment must be sufficient to make the socially desired action more attractive for the ecosystem managers [65].

- Reforestation: Reforestation is the action destined to the repopulation of zones that were previously covered by vegetation and forests and which for different reasons have been eliminated [71]. Trees incorporated into agro-ecosystems will be a fundamental part of the water cycle [71], according to the theory of the biotic pump [38,39,72]. Evapotranspiration in forested areas will increase rainfall. In addition forests are recognized as moisture sensors which act as macro regulators of the Amazon–Andes water cycle. It should also limit the "paramization" phenomena [62].

The policies of environmental education and payment for ecosystem services proposed will allow the constraining loops (B3 and B4), to become reinforcement loops (R6 and R7) targeting the progress of the Páramo ecosystems, their environmental services and inhabitants (Figure 7). Additionally, the implementation of the suggested policies will generate three new reinforcement loops (R9, R10 and R11), aimed at achieving the sustainability of these agro-ecosystems. Figure 8 also shows the effect of the reforestation policy on the collection of water in the Páramos. It generates three positive feedback loops for the water cycle (R12, R13 and R14), based on the theory of the biotic pump [38,39].

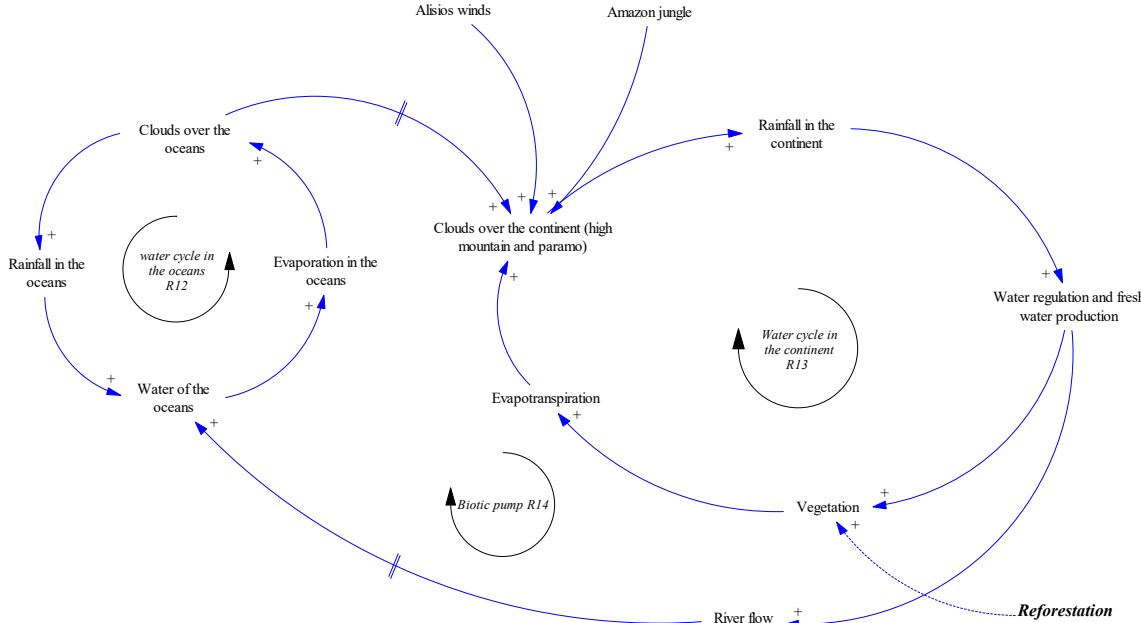

**Figure 8.** Causal loop diagram showing the effect of reforestation on the water cycle and rainfall in the Páramo area.

Finally, several of the feedback loops generated in this work through the causal diagrams are related to some of the sustainable development objectives proposed by the United Nations as Sustainable development goals (SDGs) of the Agenda 2030 (Table 2). The SDGs aim at reaching sustainable equilibrium between environmental, economic, and social dimensions of the natural resource dynamics

and human activities carried out in the Páramos. It is worth remembering that the SDGs have been supported by three pillars: (i) universality, (ii) human rights, and (iii) interconnectedness. The latter recognizes the interdependence between economic, environmental and social issues in search of actions for people, the planet, and prosperity [18]. Colombia has pioneered the inclusion of this agenda in its planning instruments such as development plans and in the national economic and social policy councils as demonstrated by the governmental law N° 1930 of 27 July of 2018. This link represents a simple association among SDGs and Páramo dynamics. It is in line with [73] in which the target stated in the 2030 Agenda of high priority across the wet landscapes are needed to achieve sustainable development: "Improved water quality", "Sustainable food production" Sustainable management of resources", and highlighted specific feedback mechanisms and synergies between SDG targets in the context of wetlands. The most consistent reinforcing interactions were the influence of sustainable food production on "Efficient resource consumption".

**Table 2.** Relationship between causal loops and sustainable development goals of Agenda 2030.

| Sustainable Development Goals | Targeted by Loop | Loop Label and Meaning |
|---|---|---|
| Goal 2. Zero Hunger. End hunger, achieve food security and improved nutrition and promote sustainable agriculture | R2 | Food security (livestock) and population growth |
| | R3 | Food security (agriculture) and population growth |
| Goal 4. Quality education. Ensure inclusive and equitable quality education and promote lifelong learning opportunities for all | R11 | Environmental education and agro-ecosystem |
| Goal 6. Clean water. Ensure availability and sustainable management of water and sanitation for all | R9 | Biotic pump |
| | R12 | Reforestation and agro-ecosystem |
| Goal 8. Economic growth and decent work. Promote sustained, inclusive and sustainable economic growth, full and productive employment and decent work for all | R4 | Livestock impact on economic income and emigration |
| | R5 | Agriculture impact on economic income and emigration |
| | B5 | Ecosystem pressure on productivity and economic income |
| Goal 13. Climate action. Take urgent action to combat climate change and its impacts | R10 | Payment for ecosystem services and agro-ecosystem |
| | R11 | Environmental education and agro-ecosystem |
| | R12 | Reforestation and agro-ecosystem |
| Goal 15. Life on land. Protect, restore, and promote sustainable use of terrestrial ecosystems, sustainably manage forests, combat desertification, and halt and reverse land degradation and halt biodiversity loss | R10 | Payment for ecosystem services and agro-ecosystem |
| | R11 | Environmental education and agro-ecosystem |
| | R12 | Reforestation and agro-ecosystem |

## 4. Conclusions

This work aimed to bring a new perspective on the understanding of Páramos as complex and dynamic systems by using the SD tools, which are very communicative and can be used by recipients with different backgrounds to boost further theoretical and practical discussions on the topic. The presented mental model provides a simplified structure of the existing relationships between the Páramo

ecological elements, their ecosystem services, and the anthropic activities of these zones. Specifically we point out that the developed CLD brings a dynamic understanding of the Páramo managerial opportunities that were not evidenced in earlier phases of the state of the art. A systems dynamics approach was applied in this study through the development of causal diagrams. It allowed to see how a holistic perspective allows testing key policies aimed at fostering sustainability and resilience of these mountainous agro-ecosystems. According to the formalized analysis, the sustainability of the Páramo is associated with the quality of anthropic activities and the human capacity to understand its own role for the ecosystem equilibrium. Expelling people from these areas to achieve conservative sustainability and then design a passive monitoring management would entail high public costs and loss of material culture for the management of the territory. Keeping the population in the area should be designed as part of the ecosystem, being the basis of the social organization, in search of mechanisms for the integral conservation of natural resources and promoting the well-being of the communities. Anthropic activities in these areas (livestock, agriculture) are currently not managed to be economically viable, socially just and environmentally responsible. Therefore, new practices of management of biodiversity and agricultural diversity must be adopted, as measures of ecosystem conservation, that generate well-being in the inhabitants. The main sustainable development goals embedded in Páramo management resulted the Goal 6 (water), 8 (economic growth, employment and work), 13 (climate change), and 15 (sustainable use of terrestrial ecosystems) summarizing the links between water and natural resources, environmental and social sustainability. To achieve this goal, it is probably necessary a state intervention aimed to modulate the incidence of the income by agricultural activities on the total incomes of the population living in the Páramos. It should be considered that the main role of the inhabitants should be the maintenance of the natural resources. Thus the integration of incomes from collateral activities and tourism paired with environmental payments should be stimulated.

**Author Contributions:** Conceptualization, Methodology and Investigation, R.A.M.B., H.S.G., L.G., R.C.G. and A.S.A.; Writing-Original Draft Preparation, R.A.M.B., A.S.A. and R.C.G.; Writing-Review & Editing, A.S.A. and R.A.M.B.

**Funding:** This research received no external funding.

**Conflicts of Interest:** The authors declare no conflict of interest.

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
