# Peer review of "Sustainable Feedbacks of Colombian Paramos Involving Livestock, Agricultural Activities, and Sustainable Development Goals of the Agenda 2030"

_systems, doi:10.3390/systems7040052_

Round 1

Reviewer 1 Report

·      In line 85 SDG is used without being spelled first.

·      Table 1 should be reformatted to the lines are not off. Somethings should be reconciled. Like why not combine line 3 and 4 since the journal, area and focus is the same. Maybe all areas would be combined? Right now, it just looks cluttered.

·      Figure 4, can you actually use the word “causal” here? Same in figure 5, etc. Could “causal” be replaced by “relationships” or “systems”? Causal loops could also be explained using a small lit review.

·      Since causal loops seems to be the methodology this needs to be furthered explain. Right now, there is only one paragraph.

·      Line 134 - are you using oxford commas or not consistently. Is seem you are not “R1, B1 and B2” vs. “R1, B1, and B2.” Just make sure you are consistent.

·      138 - is this the way this should be cited “[23]–[25]”?

·      143 - What happened to the period?

·      194 - need a space “disappearance[1]”

·      203-205 – The listing looks very awards this way. How about: 1) Environmental education, 2) Payment for ecosystem services and, 3) Reforestation

·      234 – should it be “t”?

·      Can table 2 be simplified?

·      Sentences 264-277 seem compelling, why bury this at the end of the paper. Maybe the intro could sy something like after conducing our analysis we argue that…tell me why this is important upfront. As of right now, I do not know why I am reading this, why it matters. Make me care as a reader form the beginning.

·      In the references, why the authors capitalize some of the journal titles and not others. E.g. SISTEMAS DE PRODUCCTION RURAL

Author Response

Answer to reviewers for the first submitted version of the manuscript System 550414

Sustainable feedbacks involving livestock, agricultural activities and ecosystem services of colombian paramos

REVIEWER 1

In line 85 SDG is used without being spelled first. 

Authors (AU): Thanks for the comment; acronym was explicated.

Table 1 should be reformatted to the lines are not off. Somethings should be reconciled. Like why not combine line 3 and 4 since the journal, area and focus is the same. Maybe all areas would be combined? Right now, it just looks cluttered. 

AU: Done as requested

Figure 4, can you actually use the word “causal” here? Same in figure 5, etc. Could “causal” be replaced by “relationships” or “systems”? Causal loops could also be explained using a small lit review. 

AU: The name causal loop diagrasm is the name conventionally used in the System Dynamics Methodology to identify the cuasal maps aimed to map the feedback sloops in the systems and specifically the causal relationships among variables. The name Causal loop diagram is the most appropriate for these diagrams; (See Sterman 2000. Business Dynamics. The reference book for the System Dynamics modeling). References were added in methods.

Since causal loops seems to be the methodology this needs to be furthered explain. Right now, there is only one paragraph. 

Thanks for the comment. Few lines were added to better introduce the SD methodology and some reference was also added at lines 127 -137

Line 134 - are you using oxford commas or not consistently. Is seem you are not “R1, B1 and B2” vs. “R1, B1, and B2.” Just make sure you are consistent. 

AU: Thanks, we double checked and rephrased. See lines: 176-179

138 - is this the way this should be cited “[23]–[25]”?

AU: Yes it is in the journal guidelines

143 what happened to the period? 

AU: We are sorry, the PDF version cut the line and a final point is missing the text.  See lines: 184-187

194 - need a space “disappearance[1]”

AU: Done. Thanks.

203-205 – The listing looks very awards this way. How about: 1) Environmental education, 2) Payment for ecosystem services and, 3) Reforestation 

AU: Thanks. Modified as suggested. See lines: 253-255

234 – should it be “t”?

AU: Thanks. T was erased

Can table 2 be simplified? 

AU: We added the table 2 to increase the paper readability. The only part that could be simplified are the names of the SDG, but to avoid the full explanation of the SDG we think that thos info in the table could be kept for completeness. Anyway the table 2 was edited and formatted moving the SDG in the first column and list the related loops in the second and third column o avoid repetition of the SDG.

Sentences 264-277 seem compelling, why bury this at the end of the paper. Maybe the intro could sy something like after conducing our analysis we argue that…tell me why this is important upfront. As of right now, I do not know why I am reading this, why it matters. Make me care as a reader form the beginning. 

AU: Thanks for the useful suggestion. Actually it arises from the feedback loop and causal map reading; thus it is part of the work conclusions like a take home message. Anyway we decided to make it more valuable including some idea of this in the initial part at lines…. And remark it in the conclusions.See lines: 339-361

In the references, why the authors capitalize some of the journal titles and not others. E.g. SISTEMAS DE PRODUCCTION RURAL

AU: Thanks, references were homogenized.

Reviewer 2 Report

Brief summary:

The paper attempts to provide a systemic analysis of resource use systems and their sustainability in the high mountain environment of the Colombian paramos. Based on literature review, the authors create a series of “causal loop diagrams” to map the dynamics of these systems, “the main feedbacks driving the system behaviour” (line 18), and the potential effects of selected policy interventions. The research questions are clearly indicated in lines 70-73: “[W]hich effects do the anthropogenic activities of paramo have on the sustainability of nearby agroecosystems? Which feedbacks that involve the paramos, and the nearby agroecosystems, are fundamental for the environmental sustainability and the socio-economic equilibrium in mountains and plains?”

Broad comments:

Unfortunately, the paper does not provide substantial new insights into the complex resource use systems of the paramos. A proper introduction into the study area and its ecological, socioeconomic and political context is missing, and the descriptions of the causal loop diagrams remain rather shallow and sometimes trivial (see e.g. lines 135-137: “This structure is typical of human populations; if the births (R1) are greater than the deaths and emigration of the inhabitants (B1 and B2), the population will increase, otherwise, the number of inhabitants may remain in balance or decrease [19]–[21].” Despite the claim that the paper applies a systemic approach, relations between elements are described in a simplistic and monocausal way. See, for instance, lines 137-141: “These incomers carry out agricultural activities […] which allows them to generate food for their own consumption, contributing to their independence and food security […], contributing to their nutrition and as a consequence, improving the reproduction rates of the population in a local scale”. The manifold social, cultural, political and economic factors shaping demographic changes and food security are ignored.

The paper does not contribute to the topic of the Special Issue (A Systems Approach to Sustainability within Agenda 2030 and across SDGs). The identified “feedback loops” are associated with selected SDGs (Table 2, l. 255), but the paper does not go beyond merely naming them. I do not see any added value in this.

The methods are poorly described, and several important questions are not addressed: what “Systems Dynamics methodology” (l. 106-107) do the authors draw on in what way? What criteria were applied in the selection of literature? Was the literature review done systematically? How were the causal processes and feedback loops identified from the selected literature? Did the authors apply some kind of standardized coding method to identify the main factors and relationships from the literature? Also, the selection of the three policy interventions is not sufficiently justified.

 While it does not provide significant new insights into social-ecological systems in the paramos, the paper may have the potential for a methodological contribution on qualitative system mapping/modeling based on literature review, provided that the literature research and the creation of the causal loop diagrams were done in a systematic and replicable way. For that, however, a much more detailed description of the applied methods would be necessary, as well as a discussion of the advantages and disadvantages of the approach. Furthermore, a more thorough review of literature on systems approaches would be needed.

In terms of grammar and writing style, the paper needs significant improvement.

Specific comments:

-          Abstract: avoid unscientific terms like “amazing” and “bad”

-          44-45: on what grounds represent these two aspects “the central axis”?

-     45: Tautology: "Water, vital for the management of water resources"

-          65-66: The term “sustainability” seems to be used as an empty phrase. Elaborations of the literature findings are needed!

-          73 (and other line numbers): What do the authors mean by (socio-economic) equilibrium? Needs to be defined/specified.

-          173: Why do the authors call the diagrams “mental models”? The use of this term seems arbitrary.

-          180-184: the paragraph is too general, more elaboration of causal relations needed.

-          186-194: How does this paragraph relate to the other findings?

-          259 (and elsewhere): What do the authors mean by anthropic? Anthropogenic?

-          271-273: “Anthropic activities in these areas […] do not manage to be economically viable, socially just and environmentally responsible”. Why? The paper does little to support this claim with empirical evidence.

Author Response

Answer to reviewers for the first submitted version of the manuscript System 550414

Sustainable feedbacks involving livestock, agricultural activities and ecosystem services of colombian paramos

REVIEWER 2

Brief summary: 

The paper attempts to provide a systemic analysis of resource use systems and their sustainability in the high mountain environment of the Colombian paramos. Based on literature review, the authors create a series of “causal loop diagrams” to map the dynamics of these systems, “the main feedbacks driving the system behaviour” (line 18), and the potential effects of selected policy interventions. The research questions are clearly indicated in lines 70-73: “[W]hich effects do the anthropogenic activities of paramo have on the sustainability of nearby agroecosystems? Which feedbacks that involve the paramos, and the nearby agro ecosystems, are fundamental for the environmental sustainability and the socio-economic equilibrium in mountains and plains?”

AU: we agree with this summary, thanks.

Broad comments: 

Unfortunately, the paper does not provide substantial new insights into the complex resource use systems of the paramos.

AU: we disagree with this point of view. The organization of the literature info in a big picture of the existing relationships already documented in literature is not only the evidence of new relationships but also the exhaustive connection of the components in a whole view. It allows to avoid the consideration of only narrowed links in policy formulation or to avoid successful quick solutions that could fail in the long term (Ford, A. 1999, Modelling the enviroment). Reading the single evidences from cited papers is not possible to consider the complexity but it arises when connected to everything else-

A proper introduction into the study area and its ecological, socioeconomic and political context is missing, and the descriptions of the causal loop diagrams remain rather shallow and sometimes trivial (see e.g. lines 135-137: “This structure is typical of human populations; if the births (R1) are greater than the deaths and emigration of the inhabitants (B1 and B2), the population will increase, otherwise, the number of inhabitants may remain in balance or decrease [19]–[21].”

AU:  thanks for the comment. We agree that a better description of the context could be introduced. The sentence was rephrased. At the opposite, System dynamics modeling is largely based on population dynamics and literature is very risch of discussions and argumentations among effects of birth and deaths and emigration flows. A large digression on these factors could be redundant. Anyway a sentence to relate population dynamics to local factors was included in the text to facilitate the contextualization at lines 176-180

Despite the claim that the paper applies a systemic approach, relations between elements are described in a simplistic and monocausal way. See, for instance, lines 137-141: “These incomers carry out agricultural activities […] which allows them to generate food for their own consumption, contributing to their independence and food security […], contributing to their nutrition and as a consequence, improving the reproduction rates of the population in a local scale”. The manifold social, cultural, political and economic factors shaping demographic changes and food security are ignored.

AU: the modeling have the purpose to condense the information in order to simplify the diagrams and maximize the readability. We could add hundreds of variables to make the model more inclusive and complete but loosing the easiness to get the big picture. We agreed that we also did not included the biological factors that determine agricultural and animal production (requirements, stocking rate and physical and biological variables affecting production) or market dynamics, or geological equilibrium that allows water collection etc. But this simplification did not reduced the validity of the diagram in its fundamental structure that is reported in the presented diagrams. We choose to reduce the number of variables and to keep it simple and understandable.

Any integratiuon can be furtherly added by “expanding” or “Exploiding” the model, thus changing one variable with a set of variable and their deep connections (Sterman, 2000).

The paper does not contribute to the topic of the Special Issue (A Systems Approach to Sustainability within Agenda 2030 and across SDGs). The identified “feedback loops” are associated with selected SDGs (Table 2, l. 255), but the paper does not go beyond merely naming them. I do not see any added value in this.

AU: Thanks for the comment. The relationships with the SDG were improved discussing the single relationships among the feedback loops and the SDG.at lines

The methods are poorly described, and several important questions are not addressed: what “Systems Dynamics methodology” (l. 106-107) do the authors draw on in what way? What criteria were applied in the selection of literature? Was the literature review done systematically? How were the causal processes and feedback loops identified from the selected literature? Did the authors apply some kind of standardized coding method to identify the main factors and relationships from the literature? Also, the selection of the three policy interventions is not sufficiently justified.

AU: thanks for the comment.  Method section ahs been integrated as requested.

While it does not provide significant new insights into social-ecological systems in the paramos, the paper may have the potential for a methodological contribution on qualitative system mapping/modeling based on literature review, provided that the literature research and the creation of the causal loop diagrams were done in a systematic and replicable way. For that, however, a much more detailed description of the applied methods would be necessary, as well as a discussion of the advantages and disadvantages of the approach. Furthermore, a more thorough review of literature on systems approaches would be needed.

AU: we agree, thanks for the comment. The manuscript was improved including some info on the System Dynamics approach used. However a large methodological digression might decrease the focus on the main topic. We increased the number of references since broad literature has been published on this field in the last 50 years.

In terms of grammar and writing style, the paper needs significant improvement.

AU: thanks. We improved the English as requested.

Specific comments:

Abstract: avoid unscientific terms like “amazing” and “bad”

AU: the sentences were rephrased.

44-45: on what grounds represent these two aspects “the central axis”?

AU: Good point. Thanks for the question. Firstly considering that water and organic matter are tconstitute the basis of th chemical and physical properties of the paramo substrates, as reported in literature. But moreover, the paramo’s ecosystem, especialluy for water flows and indirect effects generated by organic matter, is recognized as fundamental to the natural and anthropic ecosystems that rely at lower altitudes. See lines 46-48

45: Tautology: "Water, vital for the management of water resources"

AU: correct, the sentence was rephrased See lines 47-48

65-66: The term “sustainability” seems to be used as an empty phrase. Elaborations of the literature findings are needed!

AU: the sentence was explained and few references were added to support the meaning. See lines 66-67

73 (and other line numbers): What do the authors mean by (socio-economic) equilibrium? Needs to be defined/specified.

AU: Accepted. The sentence was rephrased. See lines 75-77

173: Why do the authors call the diagrams “mental models”? The use of this term seems arbitrary.

AU: The system thinking and system dynamics methodology assumes that causal maps or conceptual maps are the direct representation (from analogy) of the mental model developed to understand and describe the real system. In other terms mental models and conceptual maps could be used as synonyms (Sterman, 2000). Anyway, in the manuscript we decided to change some sentence and limit the use of mental model as synonym of causal lopp diagram if not justified by a specific relationship with the mental aptitude to describe the system.

180-184: the paragraph is too general, more elaboration of causal relations needed.

AU: accepted the causal relationships were elaborated and rewritten

186-194: How does this paragraph relate to the other findings?

AU: accepted. The paragraph was restructured to link it with previous sentences. See lines 236-242

259 (and elsewhere): What do the authors mean by anthropic? Anthropogenic?

Au: Accepted. Antropic > Anthropogenic

271-273: “Anthropic activities in these areas […] do not manage to be economically viable, socially just and environmentally responsible”. Why? The paper does little to support this claim with empirical evidence.

AU: accepted. The text was integrated and the sentences were supported by adequate references.

Reviewer 3 Report

I read the article with interest and the causal loop diagrams approach seems very interesting and makes it possible to construct scenarios according to "what happens if ...". I value the article that can be published in its current version. It would be very interesting to be able to quantify some variables, both of anthropic pressure and of service provision. Before paragraph 2.2 it is necessary that the description of the study area is also quantitative: how many inhabited? how much cattle? what kind of markets do you have? The CLD identifies the relationships but these relations are of the quantitative type.This operation is essential to be able to quantify the PES. Since these are government contributions, they are still PES-like and not PES core. This must be specified in the text, inserting some bibliographic references.

Author Response

Answer to reviewers for the first submitted version of the manuscript System 550414

Sustainable feedbacks involving livestock, agricultural activities and ecosystem services of colombian paramos

REVIEWER 3

I read the article with interest and the causal loop diagrams approach seems very interesting and makes it possible to construct scenarios according to "what happens if ...". I value the article that can be published in its current version. It would be very interesting to be able to quantify some variables, both of anthropic pressure and of service provision. Before paragraph 2.2 it is necessary that the description of the study area is also quantitative: how many inhabited? how much cattle? what kind of markets do you have? The CLD identifies the relationships but these relations are of the quantitative type. This operation is essential to be able to quantify the PES. Since these are government contributions, they are still PES-like and not PES core. This must be specified in the text, inserting some bibliographic references.

AU: many thanks for the comments and suggestions. Very scarce information is available for Colombian Paramos people and livestock. There are not official census that reports separate info for páramos areas. Several words were added to explain the little information available; an indicative number of people got from estimatios reported in magazine journals was also added (300’000 to 500 000 persons) to give a quantitative term in line 118

https://sostenibilidad.semana.com/medio-ambiente/articulo/paramos-que-hacer-con-la-gente-que-vive-en-los-paramos/35992

We also added some sentences relatively to the governmental contribution at lines 406-411.

Round 2

Reviewer 1 Report

There seems to be some self-plagiarism Please change the following sentences, "Drawing causal maps or CLD using ST and SD approaches is a technique for mapping the existence of the feedback within and across interacting subsystems. The modeling process demands the identification and definition of problems, the overall system conceptualization and the qualitative modeling, which often impact our initial system understanding. ST and SD can support designing better Urban Food Policies to analyze the urban food systems structure because both phases enhance the insights elicitation and increases the likelihood of performing a good policy analysis"

Author Response

There seems to be some self-plagiarism Please change the following sentences, "Drawing causal maps or CLD using ST and SD approaches is a technique for mapping the existence of the feedback within and across interacting subsystems. The modeling process demands the identification and definition of problems, the overall system conceptualization and the qualitative modeling, which often impact our initial system understanding. ST and SD can support designing better Urban Food Policies to analyze the urban food systems structure because both phases enhance the insights elicitation and increases the likelihood of performing a good policy analysis"

AU: Thanks for the comment, accepted. There was a refuse in elaborating the concept from a previous general description of the methodology. The sentence was rephrased and the concepts reelaborated.

Other changes were included along the manuscrtipt in order to improve its quality as requested by the referee. Language editings were also included.

Reviewer 2 Report

The manuscript has somewhat improved, but I still do not think it is fit for publication in Systems.
The methods section is still vague. Some crucial information is missing to make the methods replicable:
"The literature reaserch has been 150 done using published papers and other official and informal literature source using key words related with “paramo” environment and context." --> What literature database was used with what key words? How did the authors make sure not to miss any important literature? Moreover, it still remains unclear how the causal processes and feedback loops were identified, and the selection of the three policy interventions is still not justified.
The authors respond that they "choose to reduce the number of variables and to keep it simple and understandable" and that their simplification of the model "did not reduce[..] the validity of the diagram in its fundamental structure that is reported in the presented diagrams". --> If that is so, the paper needs further elaboration of their methods and criteria of selecting the "fundamental" variables and removing others.
In their response, the authors argue: "However a large methodological digression might decrease the focus on the main topic." --> I do not think a "methodological digression" is needed, but the applied methodology must be made transparent and replicable. This is not the case, seemingly because the authors did not apply a rigid methodology in building the diagrams.
The authors claim: "The relationships with the SDG were improved discussing the single relationships among the feedback loops and the SDG.at lines [line numbers missing]" --> unfortunately, there is still no substantial link to the topic of the special issue. It remains unclear why simply relating model parts to selected SDGs in table 2 is of any scientific value, and the vague sentence on SDGs in the conclusions does not help: "The main sustainable development goals which are involved in this process could be the Goal 6 (water), 8 (economic growth, employment and work), 13 (climate change) and 15 (sustainable use of terrestrial ecosystems) which describe the links among water and natural resources, environmental and social sustainability."
The paper still needs extensive editing of language and style.

Author Response

The manuscript has somewhat improved, but I still do not think it is fit for publication in Systems. The methods section is still vague. Some crucial information is missing to make the methods replicable:
"The literature reaserch has been done using published papers and other official and informal literature source using key words related with “paramo” environment and context." --> What literature database was used with what key words? How did the authors make sure not to miss any important literature? Moreover, it still remains unclear how the causal processes and feedback loops
were identified, and the selection of the three policy interventions is still not justified.

AU: Thanks for the comment and we apologize for misunderstanding the suggestion of the previous revision. We thought that a more exaustive description of the system dynamics methodology was requested. In this version of the manuscript we better defined the methodological steps followed to
built the CLD and to make the study more replicable. Method section has been implemented with details in order to clarify how the CLD were developed, defined and validated.

The authors respond that they "choose to reduce the number of variables and to keep it simple and understandable" and that their simplification of the model "did not reduce[..] the validity of the diagram in its fundamental structure that is reported in the presented diagrams". --> If that is so, the paper needs further elaboration of their methods and criteria of selecting the "fundamental"
variables and removing others.

AU: thanks for the comment, we agree. Actually, as decribed in methods, in first instance we did not built a large diagram with very detailed links and connections among paramos variables. The paramo system is a very complex environment with recongnized undefined boundaries that involve many reasearch fields (goelogy, hydrology, botanic, agriculture, antrhopology, politics, etc). Otherwise very few quantitative information are available to model and weight the relationships among variables in the studied system. In addition we found that most part of the literature focusing paramos features was very redundant in conclusions and discussions and with high level of self referential.

What we did was to develop several small CLDs that were then reduced in order to be integrated and merged in a unic diagram, aimed to summarize the multiple linear views of the paramo in a more general sistemic perspective. It would encourage the paramo management as unic entity and to consider its equilibrium fundamental for the Paramo itself and for the areas in the nearby
(lowlands) which are indirectly affected by paramo management changes. In addition we also followed adivices of Sayel (2006, quoting Forrester, 2003) which emphasizes that many applications in education and in policy design might be better handled with a collection of far simpler models. As highlighted by Sayel (2006), we would like to“avoid the common criticism that - the model is unrealistically simple- because modellers are tempted to build large and too detailed models, which often makes the situation too complicated but still unrealistic. This is not an explanation to have bad method description but to highlight that diagrams resulted from a process aimed to keep it simple starting from largely accepted assumptions and rules and not because of a poor literature research.

Otherwise we point out that the Paramos’ CLD bring a dynamic understanding of the problem that was not evidenced in other works of the state of the art. We hope this methodological clarification was enough detailed to help the publication process of this paper.

In their response, the authors argue: "However a large methodological digression might decrease the focus on the main topic." --> I do not think a "methodological digression" is needed, but the applied methodology must be made transparent and replicable. This is not the case, seemingly because the
authors did not apply a rigid methodology in building the diagrams.

AU: Thanks for the clarification we now better understand the point of the reviewer and we tried to improve the methodological description as described before and as requested.

The authors claim: "The relationships with the SDG were improved discussing the single relationships among the feedback loops and the SDG.at lines [line numbers missing]" --> unfortunately, there is still no substantial link to the topic of the special issue. It remains unclear why simply relating model parts to selected SDGs in table 2 is of any scientific value, and the vague sentence on SDGs in the conclusions does not help: "The main sustainable development goals which are involved in this process could be the Goal 6 (water), 8 (economic growth, employment and work), 13 (climate change) and 15 (sustainable use of terrestrial ecosystems) which describe the
links among water and natural resources, environmental and social sustainability."

AU: we agree; the initial idea was to simply highlight which kind of SDGs are involved in paramo management strategies because, which is simple but still innovative in the topic because in the best of our knowledge we did not find any work pointing out this link. We slightly improved the discussion on the SDGs as requested.

The paper still needs extensive editing of language and style.

AU: accepted. Many errors ere fixed and unclear sentences were rephrased; the manuscritp was largely edited in its structure, language and mispells as requested.
